# Genotype by environment interaction across water regimes in relation to cropping season response of quinoa (*Chenopodium quinoa*)

**Van Loc Nguyen**[1], **Hue Nhan Luu**[2], **Thi Hong Nhung Phan**[1], **Viet Long Nguyen**[1]\*, **Duc Ha Chu**[3], **Daniel Bertero**[4], **Néstor Curti**[5], **Peter C. McKeown**[6], **Charles Spillane**[6]\*

1 Faculty of Agronomy, Vietnam National University of Agriculture, Hanoi, Vietnam, 2 Student at Faculty of Agronomy, Vietnam National University of Agriculture, Hanoi, Vietnam, 3 Faculty of Agricultural Technology, University of Engineering and Technology, Vietnam National University Hanoi, Hanoi, Vietnam, 4 Depto, de Producción Vegetal, Facultad de Agronomía, Universidad de Buenos Aires and IFEVA-Conicet, Buenos Aires, Argentina, 5 Escuela de Agronomía, National University of Salta, Salta, Argentina, 6 Agriculture, Food Systems & Bioeconomy Research Centre, Ryan Institute, University of Galway, Galway, Ireland

\* nvlong@vnua.edu.vn (VLN); charles.spillane@nuigalway.ie (CS)

**Data Availability Statement:** All relevant data are within the manuscript.

**Funding:** This research was supported by a Vietnam Ireland Bilateral Education Exchange

## Abstract

Genotype × environment (GxE) interaction effects are one of the major challenges in identifying cultivars with stable performance across agri-environments. In this study we analysed GE interactions to identify quinoa (*Chenopodium quinoa*) cultivars with high and stable yields under different soil moisture regimes, representing control conditions, waterlogging and drought. Waterlogging and drought treatments were artificially induced using normoxia, a combination of hypoxia-normoxia, and 10% PEG (Polyethylene glycol) under hydroponic growth conditions, respectively. Both waterlogging and drought conditions significantly reduced the plant height (PH), number of leaves (NoL) and number of branches (NoB), stem diameter (SD), leaf area (LA) and dry weight (DW) of quinoa genotypes. The genotype, water regime, and genotype by water regime effects all significantly affected the measured quinoa traits. Based on the additive main effects and multiplicative interaction (AMMI) model for DW, the genotypes G18, Puno, Q4, 2-Want, Puno, Real1 x Ruy937 and Titicaca were found to exhibit tolerance and were stable across water regimes. A second-stage evaluation was conducted to test genotype × environment interaction effects in crop production field trials, selecting two contrasting seasons based on soil moisture conditions involving a diverse set of genotypes (58 varieties in total). Our results demonstrate significant variations in both growth and yield among the quinoa genotypes across the cropping seasons. The GGE analysis for grain yield indicate that field conditions matched to G × E under hydroponic experimental conditions and the cultivars G18, Q1, Q4, NL-3, G28, 42-Test, Atlas and 59-ALC were classified within a range of high productivity. Our findings provide a basis for understanding the mechanisms of wide adaptation, while identifying germplasm that enhances the water stress tolerance of quinoa cultivars at early growth stages.

(VIBE) program grant (funded by the Irish Department of Foreign Affairs (Irish Aid) to Vietnam National University of Agriculture (Vietnam) and University of Galway (Ireland) to VLN and CS, respectively. The VIBE program grant was titled "Quinoa development for food security under climate change conditions in Vietnam", Pillar 01-2022.04/VNUA 01. The funders had no role in study design, data collection and analysis, decision to publish, or preparation of the manuscript.

**Competing interests:** The authors have declared that no competing interests exist.

## Introduction

The pseudocereal crop quinoa (*Chenopodium quinoa* Willd. Amaranthaceae) is well known for its nutritional profile and environmental adaptability [1, 2] and has potential for greater contribution to global food supply and resilience challenges [1, 3]. As the pressures of climate change intensify, the urgency of developing crop varieties with enhanced yield, stability, and resistance to biotic and abiotic stressors is mounting [4, 5]. Choice of crop and variety that are well suited to different cropping locations and conditions, demands detailed understanding of genetic and environmental interactions, a domain in which quinoa is considered to show great adaptability (Gonzalez et al. 2012, Angeli and Miguel Silva 2020)). Crop adaptability requires the precise selection of genotypes for both adaptability and stability in diverse environmental conditions. Environmental parameters such as winter and spring temperature and photoperiod have emerged as major determinants of quinoa's adaptability, affecting its growth and developmental stages [6, 7]. In part because of this, it has been realized that quinoa cultivars are limited to growth in a particular range of latitudes, and that this determines their ranges of adaptability. Identifying characteristics that determine the stability of a cultivar's performance under different environmental conditions is critical. In addition, it has been found that soil moisture levels can influence quinoa's germination, growth, and yield [8], which is in contrast with its reputation as a drought-tolerant crop at later growth stages. Perhaps because of this, few studies have addressed quinoa's performance under different moisture levels under controlled and field conditions.

Genotype × environment (GxE) interactions significantly impedes the exploration of agronomic traits, including yield and its components [9, 10]. These interactions complexify genotype selection, complicating the definition of genetic contributions to observed phenotypic variations. GxE effects also pose challenges in the interpretation and generalization of genetic experimentation outcomes, as the performance of genotypes under varying environmental conditions is not consistent. In turn, this hinders the identification of genetic factors influencing key agronomic traits [10]. Previous studies revealed that quinoa exhibits high GxE effects when subjected to multi-environment trials, indicating that its performance is significantly influenced by the interaction between genotype and environmental conditions [9]. The considerable GxE effects pose challenges to the efficiency and effectiveness of quinoa breeding programs, as it complicates the selection of superior genotypes with broad adaptation (ref), due to the variable performance across different environments. Moreover, multi-environment trials serve as a critical tool in the identification of quinoa genotypes with either wide adaptation capabilities or those displaying specific adaptation to environmental conditions [9]. The utilization of multi-environment trials is critical in breeding programs, facilitating the selection of genotypes that contribute to the development of quinoa varieties with desired agronomic traits and environmental adaptability.

The evaluation of the yield performance of genotypes across environments has been carried out for the majority of major crop species, such as maize (*Zea mays*) [11, 12], rice (*Oryza sativa*) [13, 14], wheat (*Triticum aestivum*) [4, 15] and potato (*Solanum tuberosum*) [16, 17]. As with many minor and underutilised crops, there are fewer reports on the variation of phenotypical traits and the yield performance of different quinoa genotypes under different growing environments. Moreover, first stage evaluations conducted under controlled conditions need to be scaled to field conditions to ensure that the results robustly capture and characterize the Target Population of Environments (TPEs). In particular, analysis of the variation of quinoa traits under divergent soil moisture levels is required, due to its implications for enhancing crop resilience, yield consistency, and nutritional profiles. In this study, we investigated the

productivity and stability of quinoa genotypes under contrasting environments and evaluated their adaptative potential.

## Materials and methods

### Plant materials

Fifty-eight cultivated genotypes of quinoa whose originas are from different parts of the world were used (Table 1, with name and passport data included).

### Experiments under controlled and field conditions

Two experiments were run to investigate different aspects related to GxE interactions of quinoa genotypes in Vietnam. In Experiment 1, plants of each genotype in Table 1 were grown in hydroponic conditions under semi-controlled conditions in a glasshouse of the Faculty of Agronomy, Vietnam National University of Agriculture (latitude 20˚ 60'N, longitude 105˚ 56'W, altitude ~20 m.a.s.l.). To ensure a uniform rate of germination, 30 seeds of each genotype were grown in sandy soil; uniform seedlings of each genotype were selected for treatment. The seedlings were inserted through holes punched in a 7 mm-thick polystyrene board and

**Table 1. Name and passport data of quinoa genotypes.** Names in bold letters are genotypes selected for study of G × E interactions under hydroponic experiments.

| No. | Genotype | Origin | No. | Genotype | Origin |
|---|---|---|---|---|---|
| 1 | Unknown | - | 30 | G28 | Argentina |
| 2 | 2 Want*Real | Bolivia | 31 | G8 | Argentina |
| 3 | 23-GR | Chile x Bolivia | 32 | Hahui Dache | Chile |
| 4 | **2-Want** | **Bolivia** | 33 | Isluga | Chile |
| 5 | 30-Test | Chile x Bolivia | 34 | Leucaboldo | Chile |
| 6 | 37-Test | Chile x Bolivia | 35 | Linares*Lirio | US |
| 7 | 42-Test | Chile x Bolivia | 36 | NL-3 | Netherland |
| 8 | 46-Test | Chile x Bolivia | 37 | NL-6 | Netherland |
| 9 | 59-ALC | Chile x Bolivia | 38 | Palhuin | Ecuador |
| 10 | Amachuha | Bolivia | 39 | Pichaman | Peru |
| 11 | **Atlas** | **Netherland** | 40 | Pichilemu | Bolivia |
| 12 | Baer Cajon | Chile | 41 | Pison | Argentina |
| 13 | Baer I | Chile | 42 | **Puno** | **Denmark** |
| 14 | Cahuil | Chile | 43 | **Q1** | **Argentina** |
| 15 | Cande Roja | Netherland | 44 | **Q2** | **Argentina** |
| 16 | Chadmo | Chile | 45 | **Q3** | **Argentina** |
| 17 | EDK-4 | Netherland | 46 | **Q4** | **Argentina** |
| 18 | **Puno*Chadmo** | **Denmark** | 47 | **Q4bias** | **Argentina** |
| 19 | **G13** | **Argentina** | 48 | **Q5** | **Argentina** |
| 20 | **G14** | **Argentina** | 49 | **Q6** | **Argentina** |
| 21 | **G15** | **Argentina** | 50 | **Real1*Ruy 937** | **US** |
| 22 | **G17** | **Argentina** | 51 | Red | Argentina |
| 23 | **G18** | **Argentina** | 52 | Riobamba | Netherland |
| 24 | **G19** | **Argentina** | 53 | Ru-2 | United Kingdom |
| 25 | **G20** | **Argentina** | 54 | Ru-5 | United Kingdom |
| 26 | G21 | Argentina | 55 | San Miguel | Bolivia |
| 27 | G22 | Argentina | 56 | Sayaña | Boliva |
| 28 | G23 | Argentina | 57 | Surami*Ruy937 | US |
| 29 | G26 | Argentina | 58 | **Titicaca** | **Denmark** |

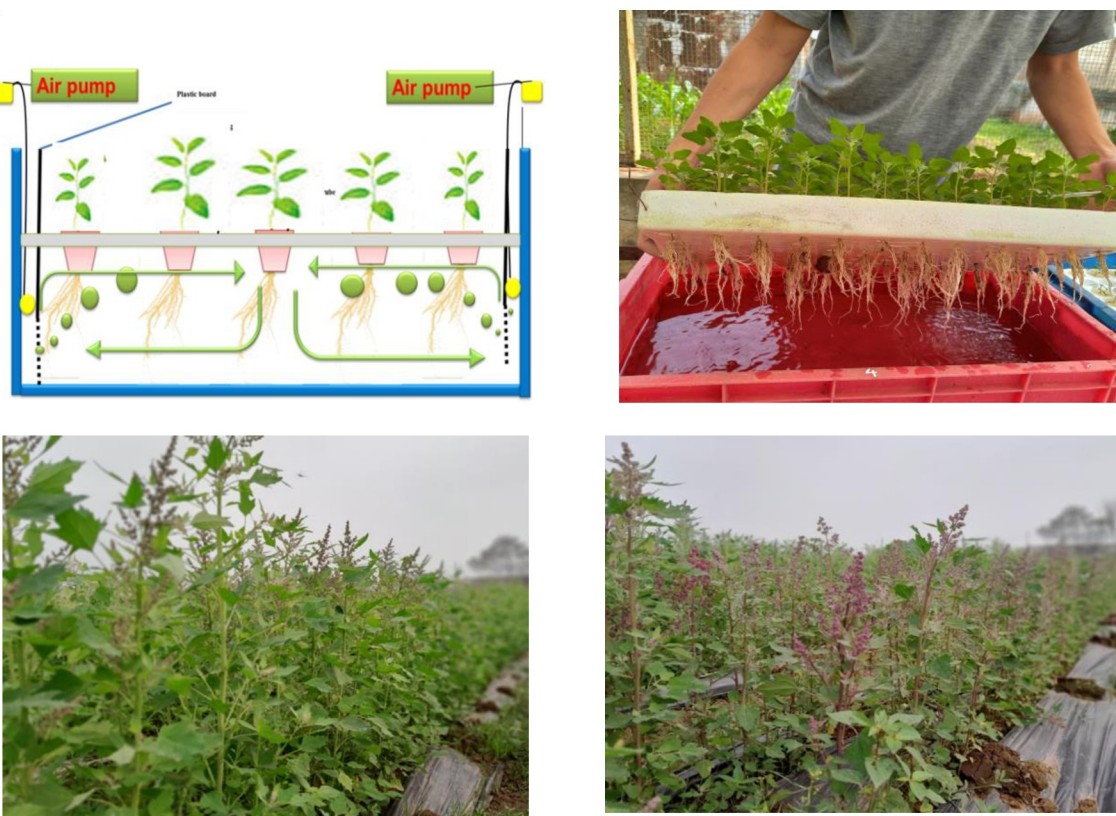

**Fig 1. A.** Hydroponic method and quinoa grown under hydroponic conditions. **B.** Quinoa genotypes grown in field trials.

held upright with plugs of silicone rubber. Each board, holding 96 seedlings, was placed over an opaque plastic container (386 mm × 256 mm × 135 mm), with the roots in deionized distilled water continuously aerated at 1.0 L min$^{-1}$ by air pumps. Plants were grown for 14 days under either waterlogging or control and drought under hydroponic culture (Fig 1A). Waterlogging and drought treatments were artificially induced using hypoxia and 10% PEG (Polyethylene glycol), respectively. Under normoxia, seedlings were grown in Kimura B solution (composed of 0.36 mMCa $(NO_3)_2 \cdot 4H_2O$, 0.36 mM $(NH_4)_2SO_4$, 0.18 mM $KH_2PO_4$, 0.18 mM $KNO_3$, 0.54 mM $MgSO_4 \cdot 7H_2O$, 40 μM Fe(III)-EDTA, 18.8 μM $H_3BO_3$, 13.4 μM $MnCl_2 \cdot 4H_2O$, 0.32 μM $CuSO_4 \cdot 5H_2O$, 0.3 μM $ZnSO_4 \cdot 4H_2O$, and 0.03 μM $(NH_4)_6Mo_7O_{24} \cdot 4H_2O$) aerated by two air pumps at the opposite ends of the container to maintain dissolved oxygen (DO) concentration of >7.0 mg L$^{-1}$. Under waterlogging, seedlings were grown in 0.1% (w/v) agar solution in deionized distilled water that is deoxygenated with a continuous flow of nitrogen to maintain a DO concentration of <1.0 mg L$^{-1}$. DO was measured with a DO meter (CM-51, Horiba, Kyoto, Japan) at the start and end of treatments. Under drought conditions, plants were grown for 14 days with 10% w/v PEG6000. To avoid drought shock injury, PEG6000 was applied stepwise to the basic nutrient solution. This began after a period of 6 hours, increasing the concentration in 5% increments on the first day of treatment. Nutrient solutions were changed every three days to avoid nutrient depletion. The experiments were arranged in a randomized complete block design with five replications (plants) per treatment. In Experiment 2, plants of each of the 58 genotypes of quinoa (Table 1) were sown at the upland field trial site of the Faculty of Agronomy, Vietnam National University of Agriculture in the winter (i.e. month 9 to month 12) and spring (i.e. month 2 to month 6) cropping seasons (Fig 1B). These

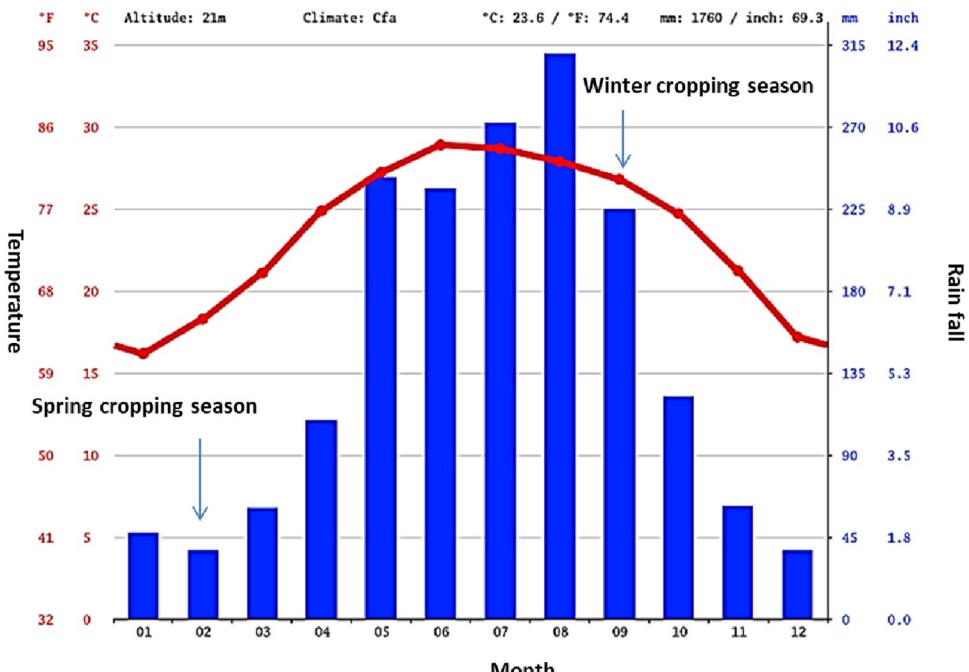

**Fig 2. Yearly climate information for Hanoi, Vietnam, including the monthly planting calendar for both the spring and winter cropping seasons.** (Source: Climate-Data.org).

two cropping seasons show contrasting conditions, especially in the early spring season when low rainfall often leads to drought stress for crops (Fig 2). Conversely, at the beginning of the winter season, relatively high rainfall can result in soil moisture excess (Fig 2). The experiments were arranged sequentially without repetition. Each experimental plot corresponding to a variety covered an area of 14 m$^2$. The cultivation procedures were carried out according to the protocol of the Vietnam National University of Agriculture as follow: 75 x 10 cm; fertilizer at a rate of 120 kg N + 90 kg P$_2$O$_5$ + 90 kg K$_2$O/ha, with 100% P$_2$O$_5$ uses as basal dressing, and nitrogen and potassium divided into three application times at 20, 40, and 60 days after emergence.

## Measurements

In experiment 1, the evaluated traits include plant height (PH), number of leaves (NoL) and number of branches (NoB), stem diameter (SD), leaf area (LA) and dry weight (DW). PH was measured from the soil surface to the top of the main stem, using 1 cm length and the presence of the axillary bud as thresholds for counting NoL and NoB, respectively. SD was measured at a 2 cm height. LA was calculated using a leaf area meter (Li-3100, Lin Coln Nebraska USA). Fresh weight was measured by using an electronic balance (OHAUS PR4202, USA). Dry weight of shoot was determined after drying samples at 80˚C for three days in a drying chamber (BINDER, USA) until a constant weight was achieved. Dry samples were weighted by the electronic balance (OHAUS STX-223, USA). Waterlogging tolerance index (WTI) and drought tolerance index (DTI) were calculated as the waterlogging and drought versus control ratio for each measured trait, respectively.

In experiment 2, the evaluated traits include growth duration (GD), PH, NoL, NoB, individual yield (IY) and theoretical yield (TY). GD was scored as days from sowing to maturity.

After harvest, theoretical yield (tonne per hectare) was calculated as individual yield multiplied by planting density.

## Data analysis

Analysis of variance (ANOVA) was performed to analyze the effects of genotype, water regimes and genotype by environment interaction using R (R Core Team, 2021). Means were cross-paired and compared using LSD at a 5% significance level in the case of significance of the impact of a factor on the measured variable. Hierarchical clustering based on principal component analysis (PCA) was performed by "factoextra" and "FactoMineR" packages in the R 4.1.3 software [18]. The additive main effects and multiplicative interaction (AMMI) and genotype main effect plus genotype by environment interaction (GGE) models were computed by "*metan*" package in R 4.1.1 [19].

## Results

Quinoa is promoted as a crop which is resilient to environmental changes, yet performance of many quinoa cultivars is affected by GxE effects. To determine the environmental stability under controlled and sub-tropical growth conditions, two experiments were carried out on a panel of 58 quinoa genotypes, first under controlled (hydroponic) conditions, then under field conditions at a field test site near Hanoi, Vietnam. Crop traits were measured and subjected to ANOVA, indicating that the genotype, water regime, and interaction between genotype and water regime had significant effects on all traits (Table 2). This confirmed that the GxE effects reported for quinoa in many parts of the world also occur under Vietnamese growth conditions. Notably, waterlogging reduced PH, NoL, NoB, SD, and LA of quinoa genotypes by 47%, 23%, 68%, 21%, and 36%, respectively, relative to the control treatment (Table 3). Drought reduced the same traits by 17%, 12%, 61%, 25%, and 31%, respectively. Among the 58 quinoa genotypes, the plant height (PH) under control condition ranged from 20.4 cm (Puno) to 50.9 cm (Q1). PH under waterlogging ranged from 12.9 cm (Q6) to 28.4 cm (G18), while PH under drought conditions ranged from 14.3 cm (Puno) to 33.7 cm (G8). WTI for PH ranged from 0.35 (Q1) to 0.91 (G18) and DTI for PH ranged from 0.64 (G15) to 0.96 (2-Want). NoL under control ranged from 18 leaves (Puno) to 24 leaves (Q4bias). NoL under waterlogging ranged from 13 (2-Want) to 18 leaves (Q4, Real1*Ruy937), while under drought conditions ranged from 13 (Q6) to 23 leaves (Q4bias). WTI for NoL ranged from 0.55 (Q6) to 0.93 (G18) and DTI for NoL ranged from 0.60 (Q6) to 1.01 (Puno). NoB under control ranged from 5 (Q2, Real1*Ruy937) to 8 branches (G15, Puno*Chadmo, Q1, Q4, and Titicaca). NoB under waterlogging ranged from one branch (2-Want, G19, Q1, Q2, Q3, Q5 and Q6) and under drought conditions also ranged from 1 branch (G14, G20, Q2, Q3, Q4, Q5, and Q6) to 5 branches

**Table 2. Mean squares and significance levels for main sources of variation resulting from ANOVA in measured quinoa plant traits.** Asterisks indicate significance level.

| Traits | Genotype | Water regime | Genotype x Water regimes |
|---|---|---|---|
| PH (cm) | 1541.01*** | 9775.73*** | 500.28*** |
| NoL (leaves plant$^{-1}$) | 48.83*** | 803.29*** | 16.59*** |
| NoB(branches plant$^{-1}$) | 135.68*** | 75.68*** | 93.65*** |
| SD (mm) | 489.33*** | 214.46*** | 130.18*** |
| LA (cm$^2$) | 4122.92*** | 15418.64*** | 988.76*** |
| DW (g plant$^{-1}$) | 725.18*** | 351.99*** | 238.88*** |

PH, plant height; NoL, number of leaves; NoB, number of branches; SD, stem diameter, LA, leaf area, DW, dry weight

Table 3. Means of measured traits at the end of stress sampling and DTI and WTI for 20 quinoa genotypes.

| Genotype | PH (cm) | | | | | NoL (leaves plant⁻¹) | | | | | NoB (branches plant⁻¹) | | | | | SD (mm) | | | | | LA (cm²) | | | | |
|---|---|---|---|---|---|---|---|---|---|---|---|---|---|---|---|---|---|---|---|---|---|---|---|---|---|
| | C | W | D | WTI | DTI | C | W | D | WTI | DTI | C | W | D | WTI | DTI | C | W | D | WTI | DTI | C | W | D | WTI | DTI |
| 2-Want | 30.2 | 15.4 | 28.9 | 0.51 | 0.96 | 20 | 13 | 20 | 0.67 | 1.00 | 6 | 1 | 5 | 0.18 | 0.74 | 3.63 | 2.45 | 3.12 | 0.67 | 0.86 | 196.78 | 95.60 | 185.34 | 0.49 | 0.94 |
| G13 | 32.5 | 15.1 | 25.1 | 0.46 | 0.77 | 22 | 16 | 19 | 0.75 | 0.88 | 7 | 3 | 4 | 0.43 | 0.55 | 3.78 | 2.78 | 2.28 | 0.74 | 0.60 | 285.74 | 126.81 | 103.24 | 0.44 | 0.36 |
| G14 | 47.6 | 17.7 | 35.6 | 0.37 | 0.75 | 19 | 15 | 16 | 0.78 | 0.87 | 6 | 2 | 1 | 0.24 | 0.11 | 4.20 | 3.25 | 2.90 | 0.77 | 0.69 | 438.00 | 165.39 | 186.28 | 0.38 | 0.43 |
| G15 | 35.3 | 17.3 | 22.4 | 0.49 | 0.64 | 19 | 15 | 17 | 0.76 | 0.89 | 8 | 2 | 5 | 0.19 | 0.60 | 3.72 | 2.70 | 2.12 | 0.73 | 0.57 | 296.24 | 104.11 | 161.26 | 0.35 | 0.54 |
| G17 | 31.1 | 12.3 | 23.3 | 0.39 | 0.75 | 21 | 15 | 15 | 0.72 | 0.73 | 6 | 4 | 3 | 0.66 | 0.49 | 2.45 | 2.35 | 1.77 | 0.96 | 0.72 | 181.15 | 86.37 | 110.68 | 0.48 | 0.61 |
| G18 | 31.3 | 28.4 | 29.4 | 0.91 | 0.94 | 19 | 17 | 18 | 0.93 | 0.95 | 6 | 4 | 5 | 0.66 | 0.83 | 3.52 | 3.12 | 3.15 | 0.89 | 0.90 | 252.12 | 198.13 | 222.30 | 0.79 | 0.88 |
| G19 | 32.3 | 16.6 | 27.8 | 0.51 | 0.86 | 20 | 16 | 16 | 0.79 | 0.80 | 7 | 1 | 2 | 0.12 | 0.27 | 3.32 | 2.13 | 1.78 | 0.64 | 0.54 | 223.35 | 113.34 | 114.97 | 0.51 | 0.51 |
| G20 | 29.7 | 17.9 | 21.4 | 0.60 | 0.72 | 21 | 16 | 17 | 0.77 | 0.81 | 6 | 3 | 1 | 0.49 | 0.14 | 2.92 | 2.13 | 1.30 | 0.73 | 0.45 | 140.57 | 75.88 | 99.87 | 0.54 | 0.71 |
| G8 | 43.7 | 19.7 | 33.7 | 0.45 | 0.77 | 21 | 15 | 20 | 0.69 | 0.94 | 6 | 2 | 2 | 0.31 | 0.31 | 3.62 | 3.02 | 2.62 | 0.83 | 0.72 | 140.68 | 120.43 | 80.08 | 0.86 | 0.57 |
| Puno | 20.4 | 18.1 | 14.3 | 0.89 | 0.70 | 18 | 17 | 19 | 0.90 | 1.01 | 7 | 3 | 4 | 0.41 | 0.56 | 3.17 | 2.92 | 2.88 | 0.92 | 0.91 | 244.47 | 188.01 | 177.54 | 0.77 | 0.73 |
| Puno*Chadmo | 26.2 | 15.5 | 23.3 | 0.59 | 0.89 | 20 | 16 | 19 | 0.81 | 0.98 | 8 | 2 | 2 | 0.23 | 0.23 | 2.38 | 1.90 | 2.08 | 0.80 | 0.87 | 70.63 | 58.47 | 65.95 | 0.83 | 0.93 |
| Q1 | 50.9 | 17.9 | 43.0 | 0.35 | 0.84 | 22 | 14 | 19 | 0.64 | 0.86 | 8 | 1 | 4 | 0.11 | 0.49 | 4.09 | 2.50 | 3.55 | 0.61 | 0.87 | 188.28 | 70.50 | 163.18 | 0.37 | 0.87 |
| Q2 | 34.7 | 18.3 | 29.7 | 0.53 | 0.86 | 19 | 15 | 17 | 0.77 | 0.89 | 5 | 1 | 1 | 0.10 | 0.17 | 2.98 | 2.68 | 2.32 | 0.90 | 0.78 | 165.20 | 130.33 | 68.28 | 0.79 | 0.41 |
| Q3 | 33.2 | 18.2 | 28.9 | 0.55 | 0.87 | 19 | 16 | 18 | 0.84 | 0.94 | 8 | 1 | 1 | 0.11 | 0.13 | 3.88 | 2.72 | 2.80 | 0.70 | 0.72 | 172.19 | 138.64 | 90.18 | 0.81 | 0.52 |
| Q4 | 33.1 | 28.5 | 30.7 | 0.86 | 0.93 | 20 | 18 | 17 | 0.88 | 0.87 | 8 | 5 | 1 | 0.62 | 0.11 | 3.98 | 3.52 | 3.30 | 0.88 | 0.83 | 193.28 | 174.10 | 177.67 | 0.90 | 0.92 |
| Q4 bias | 40.7 | 17.6 | 26.8 | 0.43 | 0.66 | 24 | 17 | 23 | 0.69 | 0.94 | 7 | 2 | 3 | 0.23 | 0.50 | 4.88 | 3.02 | 2.60 | 0.62 | 0.53 | 233.56 | 167.84 | 197.36 | 0.72 | 0.85 |
| Q5 | 36.0 | 18.5 | 33.0 | 0.51 | 0.92 | 19 | 13 | 14 | 0.70 | 0.76 | 6 | 1 | 1 | 0.14 | 0.17 | 2.83 | 2.32 | 2.80 | 0.82 | 0.99 | 211.22 | 164.34 | 168.53 | 0.78 | 0.80 |
| Q6 | 32.0 | 12.9 | 25.2 | 0.40 | 0.79 | 21 | 12 | 13 | 0.55 | 0.60 | 7 | 1 | 1 | 0.12 | 0.12 | 3.93 | 2.72 | 2.12 | 0.69 | 0.54 | 193.33 | 67.94 | 109.30 | 0.35 | 0.57 |
| Reall*Ryu937 | 32.6 | 24.7 | 30.6 | 0.76 | 0.94 | 21 | 18 | 21 | 0.86 | 1.00 | 5 | 3 | 3 | 0.59 | 0.59 | 2.82 | 2.62 | 2.42 | 0.93 | 0.86 | 149.95 | 106.87 | 128.24 | 0.71 | 0.86 |
| Titicaca | 26.5 | 20.0 | 24.8 | 0.76 | 0.94 | 19 | 17 | 18 | 0.86 | 0.92 | 8 | 3 | 5 | 0.36 | 0.62 | 3.95 | 3.52 | 3.22 | 0.89 | 0.81 | 243.15 | 201.62 | 189.68 | 0.83 | 0.78 |
| *Average* | *34.0* | *18.5* | *27.9* | *0.57* | *0.83* | *20* | *16* | *18* | *0.77* | *0.88* | *7* | *2* | *3* | *0.32* | *0.39* | *3.50* | *2.72* | *2.56* | *0.78* | *0.73* | *210.99* | *127.74* | *140.00* | *0.64* | *0.69* |

PH, plant height; NoL, number of leaves; NoB, number of branches; SD, stem diameter; LA, leaf area; C, control; W, waterlogging; D, drought; WTI, waterlogging tolerant index; DTI, drought tolerant index.

(2-Want, G15, G18, and Titicaca). WTI for NoB ranged from 0.10 (Q2) to 0.66 (G17 and G18) and DTI for NoB ranged from 0.11 (G14 and Q4) to 0.83 (G18). SD under control ranged from 2.38cm (Puno*Chadmo) to 4.88cm (Q4bias). SD under waterlogging ranged from 1.90 (Puno) to 3.52 cm (Q4) and SD under drought conditions also ranged from 1.30 (G20) to 3.30 cm (Q4). WTI for SD ranged from 0.61 (Q1) to 0.96 (G17) and DTI for SD ranged from 0.45 (G20) to 0.99 (Q5). LA under control ranged from 70.63 (Puno*Chadmo) to 438 cm$^2$ (G14). LA under waterlogging ranged from 58.47 (Puno*Chadmo) to 174.14cm$^2$ (Q4) and LA under drought conditions also ranged from 80.08 (G8) to 222.30 cm$^2$ (G18). WTI for LA ranged from 0.35 (Q1) to 0.86 (G17) and DTI for LA ranged from 0.45 (G13) to 0.94 (2-Want).

Dry weight (DW) under control conditions varied from 0.76 g (G18) to 1.51 g (G14). Under waterlogging, DW ranged from 0.32 g (G19) to 0.79 g (Real1*Ryu937), while under drought conditions, it ranged from 0.33 g (G20) to 0.91 g (Real1*Ruy937). WTI for DW ranged from 0.30 (G14) to 0.80 (G18), and DTI ranged from 0.47 (G13) to 0.88 (G18 and Titicaca) (Table 4).

To determine the stability of quinoa performance, the stability levels of DW of genotypes, and its association with test environments, were measured and represented in AMMI biplots (Figs 3 and 4). AMMI stability showing the relationship between stress tolerant quinoa genotypes and water stress environments with different water regimes is presented in 'DW vs PC1 scores' (i.e., AMMI1, Fig 3). The environment waterlogging is far from the origin with longer vectors representing strong interactions, whereas the drought environment shows shorter

**Table 4. Means of dry weight (DW) at the end of stress sampling, drought tolerant index (DTI) and waterlogging (hypoxia) tolerant index (WTI) for 20 quinoa genotypes.**

| Genotype | DW (g plant$^{-1}$) | | | | |
|---|---|---|---|---|---|
| | C | W | D | WTI | DTI |
| 2-Want | 1.00 | 0.50 | 0.87 | 0.50 | 0.86 |
| G13 | 1.28 | 0.56 | 0.61 | 0.44 | 0.47 |
| G14 | 1.51 | 0.45 | 0.70 | 0.30 | 0.46 |
| G15 | 1.05 | 0.46 | 0.70 | 0.44 | 0.67 |
| G17 | 0.79 | 0.34 | 0.52 | 0.43 | 0.65 |
| G18 | 0.76 | 0.61 | 0.66 | 0.80 | 0.88 |
| G19 | 0.80 | 0.32 | 0.50 | 0.41 | 0.62 |
| G20 | 0.63 | 0.43 | 0.33 | 0.68 | 0.51 |
| G8 | 0.81 | 0.45 | 0.57 | 0.56 | 0.71 |
| Puno | 0.98 | 0.74 | 0.77 | 0.75 | 0.79 |
| Puno*Chadmo | 0.99 | 0.44 | 0.69 | 0.44 | 0.69 |
| Q1 | 1.08 | 0.35 | 0.80 | 0.33 | 0.74 |
| Q2 | 0.89 | 0.47 | 0.61 | 0.53 | 0.69 |
| Q3 | 0.85 | 0.56 | 0.53 | 0.66 | 0.62 |
| Q4 | 0.96 | 0.75 | 0.79 | 0.78 | 0.82 |
| Q4 bias | 1.21 | 0.52 | 0.85 | 0.43 | 0.71 |
| Q5 | 0.87 | 0.34 | 0.70 | 0.39 | 0.81 |
| Q6 | 1.17 | 0.50 | 0.65 | 0.43 | 0.56 |
| Real1*Ryu937 | 1.11 | 0.79 | 0.91 | 0.72 | 0.82 |
| Titicaca | 1.07 | 0.78 | 0.94 | 0.73 | 0.88 |
| *Average* | *0.99* | *0.52* | *0.69* | *0.54* | *0.70* |

DW, dry wieght; C, control; W, waterlogging; D, drought; WTI, waterlogging tolerant index; DTI, drought tolerant index.

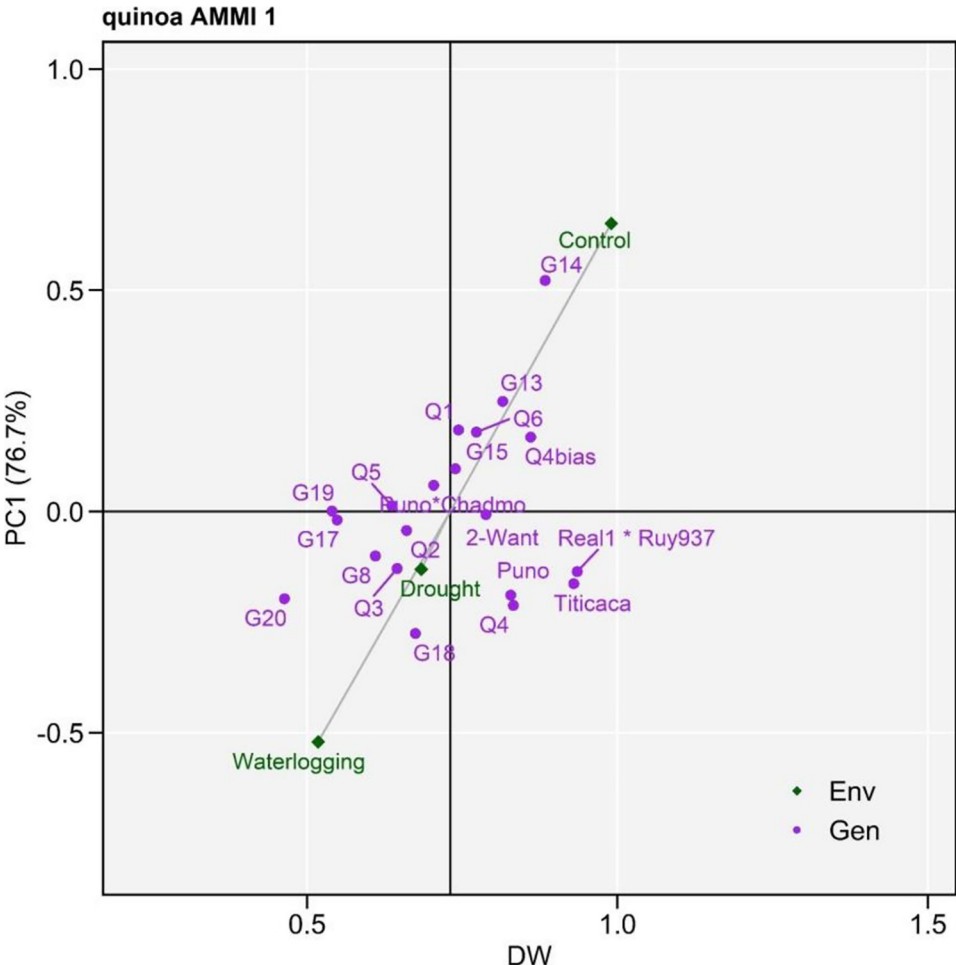

**Fig 3. Additive main effects and multiplicative interaction 1 (AMMI1) biplots based on PC1 illustrating GxE interactions of quinoa genotypes.** Env, Environment; Gen, genotype; DW, dry weight.

vectors and greater proximity to the origin, indicating weaker interaction effects. However, genotypes such as G14, Real1*Ruy937, Titicaca showed higher DW than the overall mean value. Genotypes 2-Want, Puno*Chadmo, Q5 and Q2 were placed close to the origin implying a broadly adaptation to the environments with a performance close to average overall mean yield. The AMMI2 biplot explained 100% of GxE effects of which PC1 and PC2 contributed 76.7% and 23.3% to total variation, respectively (Fig 4). The polygon view describing the vertex genotypes that are with maximum or minimum DW shown specific adaptation to the environment. A perpendicular projection from the genotype to the environmental vector revealed the amount of interaction effects with the particular environment. The genotypes G13, G14, Q1, 2-Want, G18, and G20 showed higher or lower DW and poor stable performance across environments.

This pattern allowed the visual grouping of environments based on crossed GxE effects between the high DW genotypes shown in Fig 5. Cumulative variation contributed by PC1 and PC2 was 89.7%, which suggested sufficient for fitting GGE biplot model and construction of GGE biplots [20]. A genotype that is highly stable across the environments, and also has high mean performance, is considered a more ideal genotype. The performance of a genotype in a particular environment is ranked by the axis line that passes through the center of origin.

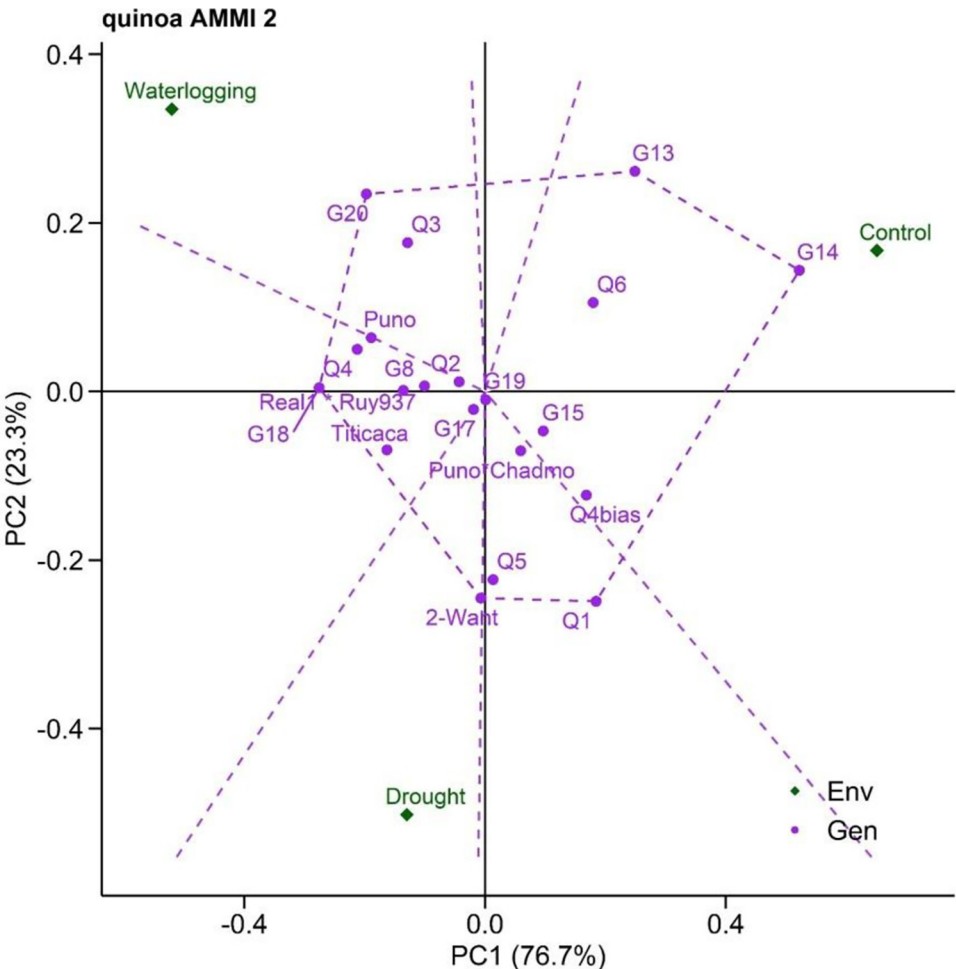

**Fig 4. Additive main effects and multiplicative interaction 2 (AMMI2) biplots based on PC1 and PC2 illustrating GxE interactions of quinoa genotypes.** Env, Environment; Gen, genotype.

An ideal genotype is mostly plotted near the center of concentric circles. It also has a vector length that is equal to the longest vector of genotypes on the positive side of the highest mean performance. The genotype Real1*Ruy937 was considered the most desirable, whereas the "G20" was considered to be the least desirable of all genotypes, as it was the furthest from the center of the concentric circle. The important findings suggested that some genotypes such as G18, Puno, Q4, 2-Want, Q4, Puno, Real1 x Ruy937, Titicaca exhibit tolerance under various water conditions (Fig 5).

To calculate overall performances of the 58 genotypes, mean values for each of the agronomic traits were calculated (Table 5). We found a large variation in many of these: GD varied from 75 to 120 days in the Spring cropping season and from 70 to 110 days in the Winter cropping season, for instance. The longest GD was recorded for genotype Q1 while the earliest variety to mature was genotype Titicaca (as might have been expected, given its Scandinavian origin). The PH ranged from 75 to 180 cm in the Spring cropping season and from 67 to 141 cm in the Winter cropping season while NoL varied from 21 to 42 leaves in the Spring cropping season and from 16 to 43 leaves in the Winter cropping season, NoB ranged from 17 to 45 branches in the Spring cropping season and from 14 to 38 branches in the Winter cropping

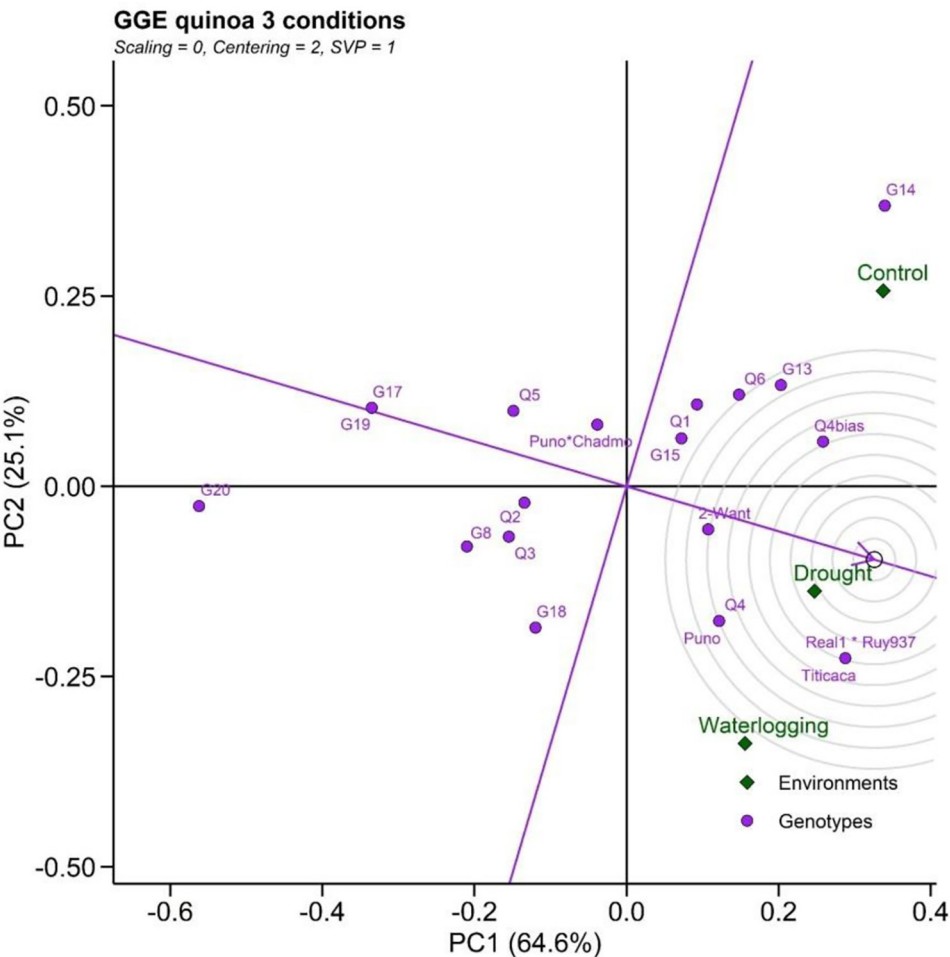

**Fig 5. GGE plot for drought and waterlogging tolerant-genotype.** The ideal genotypes plotted near the center of concentric circles.

season and IY ranged from 13 to 36 g in the Spring cropping season and from 12 to 33 g in Winter.

To relate the individual traits of cultivars into an assessment of their overall environmental stability, we performed a PCA (Fig 6). The performance of any given genotype in a particular environment can be ranked by the axis line that passes through the center of origin: an ideal genotype would be plotted near the center of concentric circles. It also has a vector length that is equal to the longest vector of genotypes on the positive side of the highest mean performance. G18 was considered the most desirable *followed* by Q1, G28, 42-Test, G28 and Atlas, while Pichilemu was considered as the poorest of all genotypes since it lay furthest from the center of the concentric circle (Fig 6). We conclude that the use of field trials to determine stability of performance traits can differentiate between quinoa cultivars which are more or less prone to GxE effects.

## Discussion

A crop genotype that is highly stable across environments and also has high mean performance is considered ideal for most agronomic purposes. Quinoa is a pseudocereal of the

**Table 5. Measured traits of quinoa genotypes under two cropping seasons.**

| Genotype | GD (days) | | PH (cm) | | NoL (leaves plant⁻¹) | | NoB (branches plant⁻¹) | | IY (g plant⁻¹) | |
|---|---|---|---|---|---|---|---|---|---|---|
| | Spring | Winter | Spring | Winter | Spring | Winter | Spring | Winter | Spring | Winter |
| Unknown | 90 | 80 | 158 | 104 | 35 | 41 | 39 | 32 | 29 | 21 |
| 2 Want*Real | 90 | 80 | 145 | 105 | 36 | 26 | 34 | 30 | 29 | 22 |
| 23-GR | 94 | 89 | 107 | 67 | 21 | 31 | 26 | 22 | 21 | 24 |
| 2-Want | 90 | 85 | 147 | 90 | 30 | 27 | 26 | 25 | 36 | 18 |
| 30-Test | 95 | 85 | 161 | 122 | 37 | 30 | 36 | 30 | 30 | 20 |
| 37-Test | 108 | 95 | 172 | 136 | 37 | 29 | 28 | 25 | 14 | 16 |
| 42-Test | 85 | 80 | 172 | 140 | 36 | 26 | 39 | 30 | 35 | 28 |
| 46-Test | 91 | 85 | 157 | 120 | 37 | 27 | 26 | 22 | 22 | 12 |
| 59-ALC | 92 | 87 | 129 | 125 | 38 | 36 | 27 | 25 | 29 | 28 |
| Amachuha | 95 | 85 | 147 | 90 | 37 | 43 | 28 | 32 | 17 | 22 |
| Atlas | 110 | 105 | 141 | 120 | 38 | 28 | 31 | 30 | 35 | 28 |
| Baer Cajon | 89 | 85 | 126 | 90 | 38 | 28 | 27 | 25 | 20 | 16 |
| Baer I | 97 | 92 | 77 | 91 | 37 | 30 | 22 | 25 | 14 | 21 |
| Cahuil | 92 | 90 | 98 | 110 | 34 | 27 | 33 | 30 | 25 | 24 |
| Cande Roja | 92 | 90 | 82 | 82 | 38 | 30 | 36 | 28 | 25 | 21 |
| Chadmo | 93 | 90 | 75 | 85 | 39 | 36 | 18 | 19 | 14 | 17 |
| EDK-4 | 100 | 87 | 144 | 105 | 38 | 28 | 29 | 27 | 25 | 15 |
| Faro*Chadmo | 90 | 80 | 172 | 105 | 33 | 28 | 13 | 14 | 23 | 17 |
| G13 | 80 | 75 | 73 | 72 | 29 | 28 | 32 | 20 | 19 | 19 |
| G14 | 85 | 75 | 156 | 83 | 33 | 30 | 24 | 17 | 20 | 13 |
| G15 | 95 | 80 | 121 | 87 | 29 | 29 | 34 | 17 | 23 | 18 |
| G17 | 95 | 80 | 119 | 80 | 30 | 28 | 17 | 20 | 19 | 19 |
| G18 | 90 | 89 | 129 | 93 | 39 | 42 | 26 | 32 | 35 | 33 |
| G19 | 90 | 75 | 122 | 82 | 31 | 21 | 23 | 17 | 18 | 13 |
| G20 | 86 | 82 | 149 | 60 | 34 | 28 | 18 | 17 | 16 | 29 |
| G21 | 98 | 85 | 158 | 120 | 38 | 34 | 27 | 27 | 16 | 21 |
| G22 | 98 | 85 | 169 | 135 | 38 | 28 | 35 | 30 | 25 | 28 |
| G23 | 90 | 80 | 163 | 120 | 36 | 29 | 31 | 30 | 18 | 23 |
| G26 | 90 | 80 | 154 | 110 | 33 | 29 | 33 | 23 | 23 | 13 |
| G28 | 98 | 90 | 158 | 120 | 37 | 39 | 34 | 28 | 33 | 31 |
| G8 | 90 | 80 | 144 | 100 | 27 | 17 | 12 | 14 | 15 | 14 |
| Hahui Dache | 92 | 87 | 156 | 126 | 37 | 26 | 24 | 21 | 25 | 20 |
| Isluga | 88 | 85 | 140 | 100 | 40 | 36 | 29 | 28 | 21 | 31 |
| Leucaboldo | 98 | 90 | 157 | 120 | 40 | 33 | 27 | 25 | 27 | 28 |
| Linares*Lirio | 95 | 90 | 111 | 91 | 37 | 27 | 26 | 22 | 21 | 23 |
| NL-3 | 92 | 87 | 76 | 76 | 34 | 34 | 25 | 20 | 31 | 31 |
| NL-6 | 90 | 85 | 125 | 115 | 37 | 41 | 39 | 29 | 25 | 22 |
| Palhuin | 95 | 90 | 133 | 83 | 36 | 35 | 33 | 31 | 27 | 21 |
| Pichaman | 92 | 85 | 103 | 83 | 32 | 42 | 23 | 21 | 20 | 26 |
| Pichilemu | 95 | 90 | 139 | 109 | 39 | 41 | 21 | 20 | 13 | 13 |
| Pison | 92 | 87 | 105 | 85 | 37 | 37 | 28 | 24 | 19 | 27 |
| Puno | 80 | 75 | 103 | 70 | 33 | 23 | 16 | 18 | 28 | 26 |
| Q1 | 120 | 110 | 175 | 140 | 38 | 43 | 32 | 30 | 35 | 33 |
| Q2 | 82 | 85 | 169 | 120 | 38 | 28 | 23 | 20 | 24 | 23 |
| Q3 | 105 | 93 | 217 | 100 | 39 | 39 | 21 | 31 | 28 | 26 |

*(Continued)*

**Table 5.** (Continued)

| Genotype | GD (days) | | PH (cm) | | NoL (leaves plant⁻¹) | | NoB (branches plant⁻¹) | | IY (g plant⁻¹) | |
|---|---|---|---|---|---|---|---|---|---|---|
| | Spring | Winter | Spring | Winter | Spring | Winter | Spring | Winter | Spring | Winter |
| Q4 | 100 | 100 | 151 | 120 | 38 | 40 | 22 | 29 | 29 | 31 |
| Q4bias | 105 | 100 | 135 | 115 | 35 | 30 | 29 | 26 | 34 | 24 |
| Q5 | 93 | 90 | 180 | 100 | 37 | 27 | 22 | 19 | 32 | 16 |
| Q6 | 83 | 75 | 126 | 87 | 26 | 16 | 17 | 18 | 30 | 20 |
| Real1*Ruy 937 | 98 | 93 | 156 | 120 | 29 | 19 | 15 | 15 | 18 | 15 |
| Red | 95 | 90 | 81 | 81 | 37 | 37 | 28 | 20 | 18 | 18 |
| Riobamba | 95 | 90 | 75 | 67 | 35 | 35 | 29 | 24 | 19 | 18 |
| Ru-2 | 95 | 93 | 134 | 141 | 40 | 31 | 45 | 27 | 26 | 22 |
| Ru-5 | 102 | 90 | 143 | 102 | 42 | 32 | 28 | 24 | 26 | 26 |
| San Miguel | 90 | 87 | 95 | 85 | 30 | 35 | 37 | 27 | 26 | 36 |
| Sayaña | 90 | 94 | 85 | 75 | 30 | 34 | 20 | 18 | 20 | 15 |
| Surami*Ruy937 | 93 | 94 | 160 | 90 | 28 | 29 | 40 | 38 | 22 | 21 |
| Titicaca | 75 | 70 | 107 | 90 | 33 | 29 | 30 | 26 | 21 | 19 |
| *Average* | *93,2* | *86,7* | *133,8* | *100,7* | *34,9* | *31,2* | *27,5* | *24,4* | *23,9* | *22,0* |
| *SD* | *7,4* | *7,4* | *32,3* | *20,6* | *4,1* | *6,3* | *7,1* | *5,4* | *6,2* | *5,9* |

GD, growth duration; PH, plant height; NoL, number of leaves (NoL); NoB, number of branches (NoB); IY individual yield; SD, standard deviation

Amaranthaceae which is promoted as a protein-rich crop with good resistance to abiotic stresses that ongoing climate change will make more frequent and intense. However, the performance of some quinoa cultivars has been found to be unreliable under conditions of variable water supply (waterlogging or drought). Under waterlogging, plants which are not adapted for aquatic conditions suffer hypoxia due to the rapid reduction of soil oxygen availability [21, 22]. Hypoxia causes multiple biochemical, cellular and physiological impacts, and over prolonged periods toxic compounds such as ethanol and acetaldehyde generated from fermentation under hypoxia stress may limit root development of susceptible cultivars [23]. Hence, excessive water can significantly inhibit crop growth and development [24].

Like many upland crops, quinoa is highly sensitive to excess moisture conditions. A study conducted in controlled growth chambers identified the adverse effects of waterlogging on the altiplano variety 'Sajama' to include: diminished plant and root dry weight, decreased total chlorophyll content, lower levels of chlorophyll a and b, and increased concentrations of soluble sugars and starch [25]. Under field conditions in Brazil, the variety 'BRS Piabiru' displayed optimal leaf measurements under a moderate water regimen. However, excessive water led to reduced leaf measurements, highlighting quinoa's vulnerability to waterlogging [26]. Most previous studies have used very limited research materials, typically only one or a few genotypes. In our study, we have significantly extended these analyses by performing a trial of 58 quinoa genotypes representative of the geographic diversity of modern quinoa cultivars to evaluate the impact of waterlogging (hypoxia) on domesticated quinoa. Our results demonstrate that quinoa as a crop is highly sensitive to waterlogging conditions, but that the extent of its susceptibility depending significantly on the genotype. Among the 20 genotypes selected for full analysis, some exhibit high tolerance, such as Real1 x Ruy937, Puno, Q4 and G18 (with HTI for DW > 0.7), indicating that resistance to waterlogging does exist within the breeding genepool of quinoa, and likely does so more extensively within the primary genepool (including landraces) of quinoa.

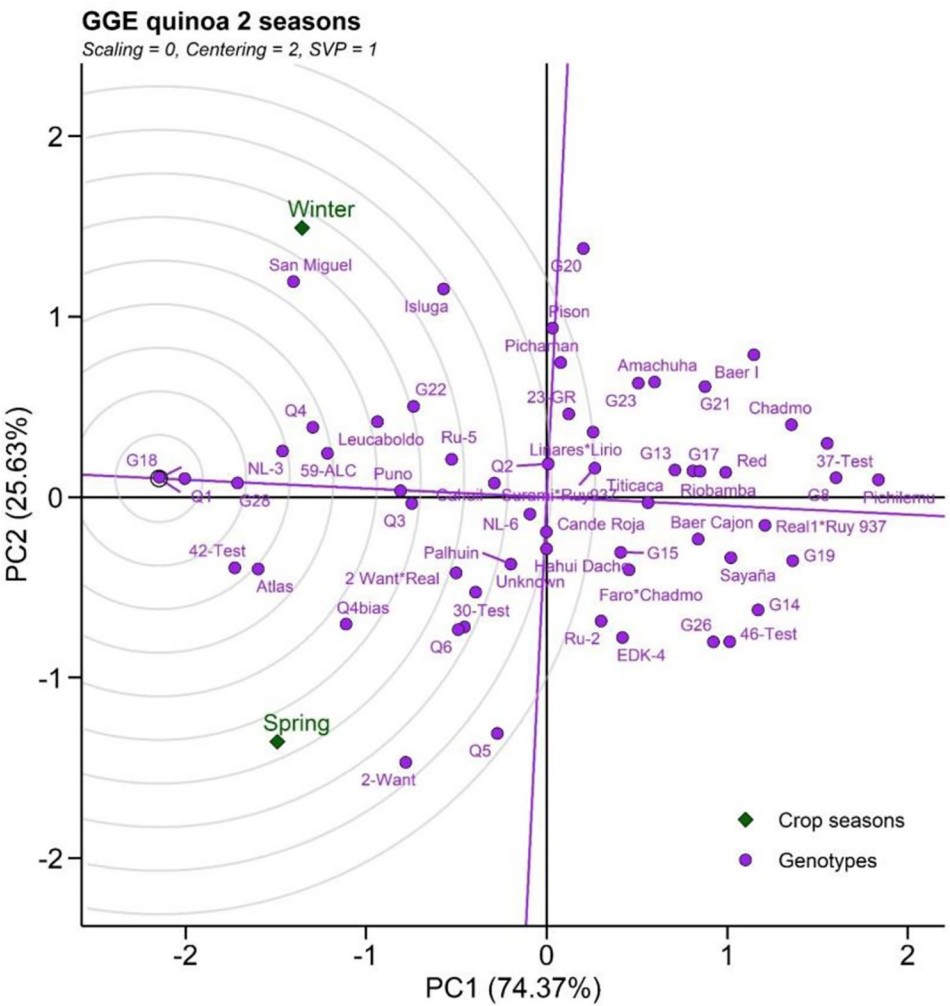

**Fig 6. GGE plots.** The ideal quinoa genotypes seasons plotted near the center of concentric circles under winter and cropping seasons.

Although quinoa is considered a drought-tolerant crop, its growth can still be significantly reduced by water deficit stress [27, 28]. Previous studies on quinoa have indicated that drought stress significantly reduces leaf area, leaf, stem and root dry weight, but does not have any significant impact on plant height [29]. Our results broadly agree with this, with reductions under our growth conditions seen in plant height, number of branches, number of leaves, shoot diameter, shoot leaf area, and dry weight (by 17%, 12%, 61%, 25%, 31%, and 30%, respectively). In addition, we observed variations in these traits among genotypes. Interestingly, we found that the accessions 2-Want, Puno, G18, Q4, Q5, Real1 x Ruy937, and Titicaca exhibited high drought tolerance (DTI for DW >0.75). In the cases of 2-Want and Puno, this confirms previous studies, but represents a novel result for the other genotypes (Iqbal et al., 2018; Nguyen et al., 2022; [6, 27, 30]. 2-Want is regarded to have arisen from a spontaneously-occurring cross between the Andean and lowland quinoa genotypes [6, 31, 32] and has improved osmotic response and antioxidant activity [27].

Cereal crops, including pseudo-cereals such as quinoa, grown in rainfed agro-ecosystems are subjected to abiotic stresses, such as low and high moisture conditions [33]. The majority

of quinoa's mega environment in the Asian tropics falls in the rainfed environment, which is extremely vulnerable to climate change [34]. The complexity of GxE effects can only be analysed using robust statistical approaches, which allow the identification of stable, adoptable genotypes for deployment. The nature of these models may vary between crop types, but all ultimately aim to identify stress-resilient genotypes of a given crop for use in any target environment. In this study, we have identified significant variation in the magnitude of environmental, genotypic and GxE effects across the quinoa cultivars and conditions studied, while also finding that overall performance under each of the target environments diverges greatly. From our analyses, highly optimal genotypes for performance across the different soil moisture regimes are identified from combined ANOVA and PCA from AMMI and GGE biplots, which indicated the existence of a crossover interaction in the responses of the cultivars., This allowed a ranking of the genotypes, with the most optimal appearing near the center of concentric circles on the plot (Fig 6). Our findings indicate that genotypes such as G18, Puno, Q4, 2-Want, Q4, Puno, Real1 x Ruy937, Titicaca exhibit tolerance or performance across various water conditions and should be prioritized in future field trials or as progenitor material in breeding programmes geared towards climate-resilient varieties for Vietnam and neighbouring countries in the region.

In addition, to quantify the GxE interaction under production conditions, we selected two contrasting seasons in terms of soil moisture conditions (at the seedling stage) for validation (experiment 2). Our results show significant variation in the growth and yield of quinoa genotypes in the two cropping seasons, with some genotypes suitable for one or other season but others displaying adaptability across both (confirming the variation observed in experiment 1). The G18, Q4, and Puno genotypes with HTI and DTI > 0.75 are all classified within the range of high productivity for both cropping seasons. Recovery capacity after stress is a critical aspect of abiotic stress tolerance [35] and its consequences are seen on seed yield and total biomass [6]. A major component of crop recovery is cropping duration (wherein a longer-cycle genotype will have more time to recover from stress). In our study, cropping durations were longer in Atlas and Q1 (>110 days in two cropping seasons) suggesting that their different sensitivities to stress could be related to increased recovery time. Both varieties were also selected as suitable for both cropping seasons, with particularly low WTI observed in the controlled experiment for Q1. G18, Q1, Q4, NL-3, G28, 42-Test, Atlas and 59-ALC are all classified within the range of high productivity for both cropping seasons. Further study of these lines can faciliatte the identification of the mechanisms of adaptation to water stress tolerance, especially during the establishment phase, in quinoa cultivars.

Overall, our findings identify prioritized germplasm and multi-environment approaches for future quinoa breeding programs, for defining mechanisms of adaptation to different water availability conditions (including under climate change impacts) and a route towards varieties which maintain their yield characters under future climate disruptions in Vietnam and South-East Asia.

## Acknowledgments

We thank to Dr. Robert Van Loo Wageningen UR for providing some genotypes for this study.

## Author Contributions

**Conceptualization:** Van Loc Nguyen, Viet Long Nguyen, Daniel Bertero, Charles Spillane.

**Data curation:** Van Loc Nguyen, Hue Nhan Luu, Thi Hong Nhung Phan, Néstor Curti.

**Formal analysis:** Hue Nhan Luu, Thi Hong Nhung Phan, Néstor Curti.

**Funding acquisition:** Van Loc Nguyen, Charles Spillane.

**Methodology:** Van Loc Nguyen, Hue Nhan Luu, Viet Long Nguyen, Daniel Bertero, Charles Spillane.

**Software:** Thi Hong Nhung Phan, Néstor Curti.

**Supervision:** Van Loc Nguyen, Viet Long Nguyen, Daniel Bertero, Peter C. McKeown, Charles Spillane.

**Writing – original draft:** Van Loc Nguyen, Duc Ha Chu, Peter C. McKeown.

**Writing – review & editing:** Van Loc Nguyen, Viet Long Nguyen, Néstor Curti, Peter C. McKeown, Charles Spillane.

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
