## [Decision Letter · Decision Letter 0]

6 Aug 2024

PONE-D-24-17742Genotype by environment interaction across water regimes in relation to cropping season response of quinoa (Chenopodium quinoa)PLOS ONE

Dear Dr. Van Loc,

Thank you for submitting your manuscript to PLOS ONE. After careful consideration, we feel that it has merit but does not fully meet PLOS ONE’s publication criteria as it currently stands. Therefore, we invite you to submit a revised version of the manuscript that addresses the points raised during the review process.

We look forward to receiving your revised manuscript.

Kind regards,

Didier Bazile

Academic Editor

PLOS ONE

Journal Requirements:

"This research was supported by a Vietnam Ireland Bilateral Education Exchange (VIBE) program grant (funded by the Irish Department of Foreign Affairs (Irish Aid) to Vietnam National University of Agriculture (Vietnam) and University of Galway (Ireland) to VLN and CS, respectively. The VIBE program grant was titled “Quinoa development for food security under climate change conditions in Vietnam”, Pillar 01-2022.04/VNUA 01."

Additional Editor Comments:

The paper is well written and the statistics are correct.

The topic is of interest for the journal.

Please read carefully of the comments of the reviewers for proceding to your revision of the paper.

Reviewers' comments:

Reviewer's Responses to Questions

**Comments to the Author**

1. Is the manuscript technically sound, and do the data support the conclusions?

Reviewer #1: Yes

Reviewer #2: Yes

2. Has the statistical analysis been performed appropriately and rigorously? 

Reviewer #1: Yes

Reviewer #2: Yes

3. Have the authors made all data underlying the findings in their manuscript fully available?

Reviewer #1: Yes

Reviewer #2: Yes

4. Is the manuscript presented in an intelligible fashion and written in standard English?

Reviewer #1: Yes

Reviewer #2: Yes

5. Review Comments to the Author

Reviewer #1: Below are comments and questions for the manuscript

The research is informative since it investigated on a large number of genotypes.

- The hydroponic growth should be mentioned for experiment 1. That will help readers can link to the the term hydroponic at the end part of the abstract.

- Does the G18 has the same response between two two experiments (basing on GGE plots)?

- Plants were treated (waterlogging, drought) immediately when seedlings were transferred to the hydroponic growth?

- Line 101: English expressions, each genotype in Table 1 was grown... instead of "plants of each of the genotypes".

- Table 1: Why select those 20 genotypes for hydroponic experiment?

- The calculation of WTI and DTI should be explained clearer (line 138-140). Were they calculated by divided traits under treatments (waterlogging or drought) by under control?

- Leaf area was measured for total leaves? or a specific leaf type?

- Term should be used consistently (Table 4. Hypoxia tolerance index – HTI. This does not present in the table)

- What is “test environments” (line 198)?

- Why PC1 and PC2 make up 100% (line 205)? Normally less than 100%

- Line 207: not grain yield?

- GGE plots did not show similar response between hydroponic and field growth (i.e., G18)?

- Line 273-274: does not make sense since drought tolerance does not cover all abiotic stressors.

Reviewer #2: The paper is directed on quinoa, a novel crop that shall increase its importance worldwide. It is well written; the statistics is correct and provide basic information to support discussion and conclusions. Given this, it is worth publishing. However, I suggest it be submitted to a more specialized journal, with focus on genetics and breeding. In this way it may reach the public involved in crop improvement.

6. PLOS authors have the option to publish the peer review history of their article (what does this mean?). If published, this will include your full peer review and any attached files.

Reviewer #1: **Yes: **Van Lam Nguyen

Reviewer #2: **Yes: **I am ready to provide additional information, whenever required os asked.

---

## [Author Response · Author response to Decision Letter 0]

9 Aug 2024

August 6th 2024

Dear Editor of Plos One

Dear Reviewers

Firstly, let us give our gratitude for your valuable recommendations

Following your recommendations and the reviewer’s comments, we have revised the manuscript accordingly (revisions are highlighted in track change). Below are our responses to reviewer´s specific comments. 

Thanks so much again for your support !

Sincerely yours,

Authors,

Authors’ revision for comments

Reviewer 1:

Thank you very much for appreciating the value of our manuscript with minor comments

Below are our responses to your specific comments.

Comment #1: The hydroponic growth should be mentioned for experiment 1. That will help readers can link to the the term hydroponic at the end part of the abstract.

Authors’ response for comment #1: Thank you very much for your comment. We’ve mentioned hydroponic growth for experiment 1 in Abstract accordingly:

Waterlogging and drought treatments were artificially induced using normoxia, a combination of hypoxia-normoxia, and 10% PEG (Polyethylene glycol) under hydroponic growth conditions, respectively

Comment #2: Does the G18 has the same response between two two experiments (basing on GGE plots)?

Authors’ response for comment #2: Thank you very much for your comment. We found that G18 is a good material, showing water stress tolerance under experiments, including field trials which will be discussed in our next paper

Comment #3: Plants were treated (waterlogging, drought) immediately when seedlings were transferred to the hydroponic growth?

Authors’ response for comment #3: Thank you very much for your question. This is to clarify the experimental method. We’ve checked and added information about the drought stress treatments (no need to adjust the shock due to oxygen deficiency because the oxygen reduction treatment simulates real conditions) in the Materials and Methods section accordingly:

Under drought conditions, plants were grown for 14 days with 10% w/v PEG6000. To avoid drought shock injury, PEG6000 was applied stepwise to the basic nutrient solution. This began after a period of 6 hours, increasing the concentration in 5% increments on the first day of treatment

Comment #4: Line 101: English expressions, each genotype in Table 1 was grown... instead of "plants of each of the genotypes".

Authors’ response for comment #4: Thank you very much for your comments. We’ve revised accordingly

Comment #5: Table 1: Why select those 20 genotypes for hydroponic experiment?

Authors’ response for comment #5: Thank you very much for your comments. We expected to proceed with all the materials; however, due to the volume of the experiment, hydroponic culture is complex and requires substantial labor and specific experimental conditions. Therefore, we have chosen 20 genotypes with diverse origins to use.

Comment #6: The calculation of WTI and DTI should be explained clearer (line 138-140). Were they calculated by divided traits under treatments (waterlogging or drought) by under control?

Authors’ response for comment #6: Thank you very much for your comments. We’ve checked, disscussed and thought that the sentence “Waterlogging tolerance index (WTI) and drought tolerance index (DTI) were calculated as the waterlogging and drought versus control ratio for each measured trait, respectively” is clear enough.

Comment #7: Leaf area was measured for total leaves? or a specific leaf type?

Authors’ response for comment #7: Thank you very much for your comments, leaf area was measured for total leaves

Comment #8: Term should be used consistently (Table 4. Hypoxia tolerance index – HTI. This does not present in the table)

Authors’ response for comment #8: Thank you very much for your comments. HTI was our mistake, we’ve changed HTI to WTI. 

Comment #8: What is “test environments” (line 198)?

Authors’ response for comment #8: Thank you very much for your comments, we’ve revised test environments to water stress invironments

Comment #9: Why PC1 and PC2 make up 100% (line 205)? Normally less than 100%

Authors’ response for comment #9: Thank you very much for your comments. We have checked and explained that we ran principal components (PCs) for a single trait of genotypes under three environments so that PC1 and PC2 account for 100% of the variation. This approach is consistent with all previous research

Comment #10: Line 207: not grain yield?

Authors’ response for comment #10: Thank you very much for your comments. We have checked and revised accordingly.

Comment #11: GGE plots did not show similar response between hydroponic and field growth (i.e., G18)?

Authors’ response for comment #11: Thank you very much for your comments. We agree with you, as field growth was influenced by other environmental factors.

Comment #12: Line 273-274: does not make sense since drought tolerance does not cover all abiotic stressors.

Authors’ response for comment #12: Thank you very much for your comments. We have checked and revised accordingly: Although quinoa is considered a drought-tolerant crop, its growth can still be significantly reduced by water deficit stress.

Reviewer 2: The paper is directed on quinoa, a novel crop that shall increase its importance worldwide. It is well written; the statistics is correct and provide basic information to support discussion and conclusions. Given this, it is worth publishing. However, I suggest it be submitted to a more specialized journal, with focus on genetics and breeding. In this way it may reach the public involved in crop improvement.

Authors’ response for comment: Thank you very much for appreciating the value of our manuscript with recommendation for specialized journal. We will consider for our next papers.

Journal comments:

We've checked your submission and before we can proceed, we need you to address the following issues:

1.In your Methods section, please provide additional information regarding the permits you obtained for the work. Please ensure you have included the full name of the authority that approved the field site access and, if no permits were required, a brief statement explaining why.

Authors’ response for comment: 

We have added the information on field site acess: . In Experiment 1, plants of each genotype in Table 1 were grown in hydroponic conditions under semi-controlled conditions in a glasshouse of the Faculty of Agronomy, Vietnam National University of Agriculture (latitude 20o 60’N, longitude 105o 56’W, altitude ~20 m.a.s.l.). In Experiment 2, plants of each of the 58 genotypes of quinoa (Table 1) were sown at the upland field trial site of the Faculty of Agronomy, Vietnam National University of Agriculture in the winter (i.e. month 9 to month 12) and spring (i.e. month 2 to month 6) cropping seasons.

 The research site and facilities belong to the Vietnam National University of Agriculture, with staff assigned to these sites and facilities.

2.Thank you for stating the following financial disclosure:"This research was supported by a Vietnam Ireland Bilateral Education Exchange (VIBE) program grant (funded by the Irish Department of Foreign Affairs (Irish Aid) to Vietnam National University of Agriculture (Vietnam) and University of Galway (Ireland) to VLN and CS, respectively. The VIBE program grant was titled “Quinoa development for food security under climate change conditions in Vietnam”, Pillar 01-2022.04/VNUA 01."

Please state what role the funders took in the study.

If the funders had no role, please state:

Please include this amended Role of Funder statement in your cover letter;

we will change the online submission form on your behalf.

Authors’ response for comment: we stated "The funders had no role in study design, data collection and analysis, decision to publish, or preparation of the manuscript” in cover letter

3.Please include your full ethics statement in the ‘Methods’ section of your manuscript file. In your statement, please include the full name of the IRB or ethics committee who approved or waived your study, as well as whether or not you obtained informed written or verbal consent. If consent was waived for your study, please include this information in your statement as well.

Authors’ response for comment: We added accordingly in “Methods”: This research was supported by a Vietnam Ireland Bilateral Education Exchange (VIBE) program grant (funded by the Irish Department of Foreign Affairs (Irish Aid) to Vietnam National University of Agriculture (Vietnam) and University of Galway (Ireland). The VIBE program grant was titled “Quinoa development for food security under climate change conditions in Vietnam”, Pillar 01-2022.04/VNUA 01.

4."In the online submission form, you indicated that [Data are available from the Institutional Data Access for researchers who meet the criteria for access to confidential data.].

3. Uploaded as supplementary information.

This policy applies to all data except where public deposition would breach compliance with the protocol approved by your research ethics board. If your data cannot be made publicly available for ethical or legal reasons (e.g., public availability would compromise patient privacy), please explain your reasons on resubmission and your exemption request will be escalated for approval."

Authors’ response for comment: We’ve added accordingly

5."Please amend your list of authors on the manuscript to ensure that each author is linked to an affiliation.

We note that you have included affiliation numbers 1,2,3,4,5 and 6 however only affiliations 1,3,4,5 and 6 have authors linked to them. Please amend affiliation 2 to link an author to it or remove if added in error.

Authors’ response for comment: We’ve revised accordingly

---

## [Editor Report · Decision Letter 1]

20 Aug 2024

Genotype by environment interaction across water regimes in relation to cropping season response of quinoa (Chenopodium quinoa)

PONE-D-24-17742R1

Dear Dr. Van Loc,

We’re pleased to inform you that your manuscript has been judged scientifically suitable for publication and will be formally accepted for publication once it meets all outstanding technical requirements.

Kind regards,

Didier Bazile

Academic Editor

PLOS ONE
---

## [Editor Report · Acceptance letter]

23 Aug 2024

PONE-D-24-17742R1 

PLOS ONE

Dear Dr. Loc Nguyen, 

I'm pleased to inform you that your manuscript has been deemed suitable for publication in PLOS ONE. Congratulations! Your manuscript is now being handed over to our production team.

Kind regards, 

on behalf of

Dr. Didier Bazile 

Academic Editor

PLOS ONE